# Digital Twins Collaboration for Automatic Erratic Operational Data Detection in Industry 4.0

**Radhya Sahal** [1,2,*], **Saeed H. Alsamhi** [3,4], **John G. Breslin** [1], **Kenneth N. Brown** [2] and **Muhammad Intizar Ali** [5]

1   Confirm SFI Research Centre for Smart Manufacturing, National University of Ireland Galway,
    H91 TK33 Galway, Ireland; john.breslin@nuigalway.ie
2   School of Computer Science and Information Technology, University College Cork, T12 E8YV Cork, Ireland;
    k.brown@cs.ucc.ie
3   Software Research Institute, Athlone Institute of Technology, N37 W089 Athlone, Ireland; salsamhi@ait.ie
4   Faculty of Engineering, IBB University, Ibb 70270, Yemen
5   School of Electronic Engineering, Dublin City University, D09 V209 Dublin, Ireland; ali.intizar@dcu.ie
*   Correspondence: radhya.sahal@nuigalway.ie

**Abstract:** Digital twin (DT) plays a pivotal role in the vision of Industry 4.0. The idea is that the real product and its virtual counterpart are twins that travel a parallel journey from design and development to production and service life. The intelligence that comes from DTs' operational data supports the interactions between the DTs to pave the way for the cyber-physical integration of smart manufacturing. This paper presents a conceptual framework for digital twins collaboration to provide an auto-detection of erratic operational data by utilizing operational data intelligence in the manufacturing systems. The proposed framework provide an interaction mechanism to understand the DT status, interact with other DTs, learn from each other DTs, and share common semantic knowledge. In addition, it can detect the anomalies and understand the overall picture and conditions of the operational environments. Furthermore, the proposed framework is described in the workflow model, which breaks down into four phases: information extraction, change detection, synchronization, and notification. A use case of Energy 4.0 fault diagnosis for wind turbines is described to present the use of the proposed framework and DTs collaboration to identify and diagnose the potential failure, e.g., malfunctioning nodes within the energy industry.

**Keywords:** digital twins; auto-detection; operational data; cyber-physical; Industry 4.0; production system

## 1. Introduction

The concept of using "Twins" originates from NASA's Apollo program (National Aeronautics and Space Administration), where at least two identical spacecrafts were built to reflect the conditions of the spacecraft during a mission in the outer space [1]. Then, several definitions of digital twins (DTs) were launched in the industrial markets. Based on literature research, we adopt the definition that the DT is an exact digital replica of something in the physical world. The DTs data is the data gathered from the physical world, which is sent by the Internet of things (IoT) sensors. Accordingly, a DT representation of a physical asset has the following features [2]: (1) the DT representation has realistic models and all available operational data on the physical asset; (2) the data has to contain all process data acquired during operation, as well as all organizational and technical information created during the development of the asset and production; (3) the DT has to be always in sync with the physical asset; and (4) it has to be possible to simulate the DT of the behavior of the physical asset.

To acquire the benefits of the DTs in manufacturing, the deployed DTs are interacting to automatically detect the erratic operational data by understanding the DT status, interacting

with other DTs, learning from other DTs, sharing common knowledge, detecting the anomalies at the edge level, and understanding the overall picture and conditions of the operational manufacturing environments. The DT receives data from wind turbines sensors in real-time, offering the status situation, future failures and detecting under performances. DTs' collaboration of wind turbines reduced maintenance costs by reducing visits to the wind farm and fixed failure. It replaced, predicting component fail time, reasons, fixed and providing failure position. Using DTs collaboration on wind farms, the energy will improve, creating renewable wind energy more affordable and help Industry 4.0 to become sustainable, more intelligent and greener.

In Figure 1, we have described the high-level of the automatic detection framework of DTs (i.e., sensors data) using semantic models and operational data (i.e., failure and erroneous data). It is shown that the consuming operational data from DTs is analyzed at edge level based on semantic models, including knowledgebase management [3]. The analysis identifies whether the DT data is a failure or erratic and shows whether the problem is happening locally or happening for all adjacent DTs representing the same device. In case there is a problem that exists for all of the deployed DTs, the system can intelligently react by sending an early notification about the failure to the decision-makers. Otherwise, the system ignores the erroneous data extracted from that DT and then updates it with the correct data to keep the operational data consistent within the production system.

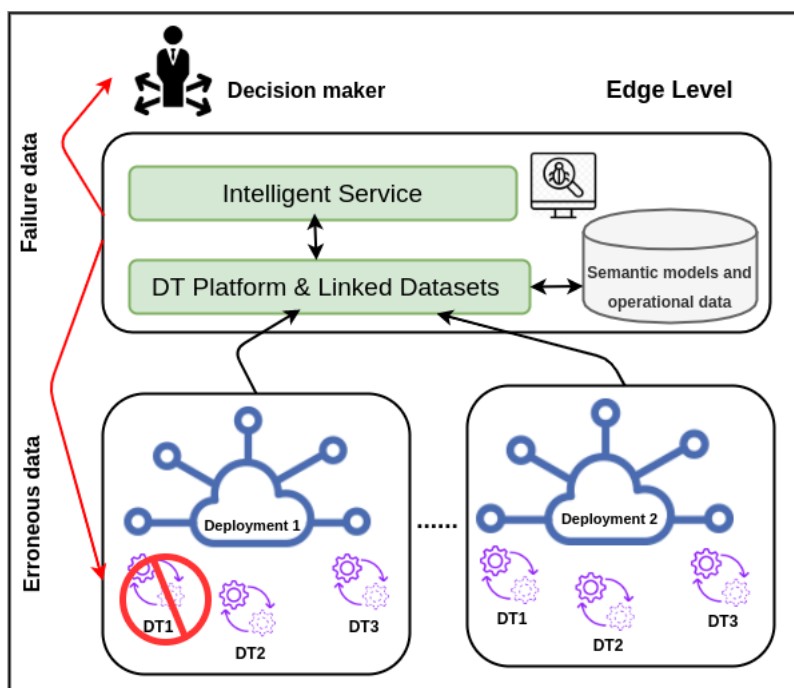

**Figure 1.** The high-level of the automatic detection of collaborative digital twins (DTs) using semantic models and operational data (i.e., failure and erroneous data).

In order to guide this research work, we stated these two research questions:

- (RQ1) *Is it possible to automatically detect DTs (sensors/device) producing failure/erroneous observations at a high level of abstraction within collaborative DTs?*
- (RQ2) *In case the observations are failures, is this failure detected by one DT or by all deployed DTs in the production?*

To answer these questions, we proposed a conceptual framework for collaborative DTs to auto-detect the erratic operational data to answer these questions. The proposed framework consists of six components: intelligent services, operational data management, knowledgebase management, models, synchronization, and simulation. The inputs will be DTs data which are the observations produced by the sensors within the production

system (i.e., the data is also represented virtually using the deployed DTs). The outputs will be indicating whether the inputs data are considered failures or erroneous. This output will be published as semantic information so that the decision-makers can take advantage of it by receiving an early notification about the production system's failure. In addition, this semantic information could be stored within the knowledgebase to be used for future auto-detection of the sensors in the cyber-physical production system. Furthermore, the proposed conceptual framework's workflow will be presented; and it consists mainly of four phases: information extraction, change detection, synchronization, and notification.

### 1.1. Contribution

To the best of our knowledge, there is no auto-detection framework for collaborative DTs that have been proposed to show how collaborative DTs and intelligence of operational data improve the smartness of the cyber-physical production system. In particular, the proposed conceptual framework collaborative DTs' auto-detection can detect whether the DT has erratic behavior or not by interacting with other collaborative DTs within the edge-level. It focuses mainly on the interaction between the deployed DTs to automatically detect the erratic DT data by understanding the DT status, interacting with other DTs, learning from each other DTs, and sharing common semantic knowledge. Our main contributions in this conceptual framework paper can be summarized as follows:

- A conceptual framework using collaborative DTs provides an interaction mechanism for auto-detecting the erratic operational data that can understand the DT status, interact with other DTs, learn from other DTs, and share common semantic knowledge.
- A workflow description of the DTs interaction which consists mainly of four phases: information extraction, change detection, synchronization, and notification.
- Energy 4.0 fault diagnosis of wind turbines use case.

### 1.2. Paper Organization

The remainder of this paper is organized as follows: The related work is introduced in Section 2. An overview of DTs collaboration is introduced in Section 3. The proposed conceptual framework for collaborative DTs is introduced in Section 4. The workflow of the proposed framework is presented in Section 5. The use case of the fault diagnosis for the wind energy industry is described in Section 6. The discussion and future direction are presented in Section 7. Finally, conclusions are presented in Section 8.

## 2. Related Work

Collaboration is vital to perform complex tasks efficiently and collaboratively, while a single user cannot do it [4]. The authors of Reference [5] discussed homogeneous and heterogeneous multi-robot collaboration to perform their complex task in a decentralized fashion with blockchain technology. The authors of Reference [6] addressed the importance of machine learning of multi-robot cooperation by keeping connectivity and improving mobility in performing tasks. The authors of Reference [7] discussed the collaboration between drone and IoT devices for improving Industry 4.0 applications, such as smart city, smart healthcare, etc. However, not one of the above studies discussed how DTs can collaborate to improve Industry 4.0 application. However, we discussed the collaboration concept in DTs based on the above studies, and we discussed wind turbines as used case for Energy 4.0 of Industry 4.0.

Smart manufacturing depends on the vast amounts of data collected by sensors and/or their DTs and the production line's data. These data could be erratic sensors and/or their DTs producing and injecting incorrect data in analysis, affecting decision-making. We will present a brief review of the state-of-the-art studies considering two main research areas: the role of DT for both data modeling and data analytics in manufacturing systems.

Wind power 4.0 is one of the renewable energy technology that requires monitoring, delivering, and analyzing. The authors of Reference [8] presented DT monitoring that can

be used for monitoring and development of wind farms. For fault diagnosis, the authors of Reference [9] discussed an approach to identify the network physical Vulnerabilities in Industry 4.0 systems systematically. It shows the potential impact of identifying vulnerabilities in Industry 4.0. Moreover, the authors of Reference [10] introduced supervisory control and data acquisition method for prediction and fault diagnosis of wind turbines using artificial intelligence. By gathering and analyzing big data from smart manufacturing products, it could take the initiative to determine the products fault diagnosis and health monitoring [11].

Concerning DT and data modeling, many research works use semantic technologies to enhance DTs models. Evgeny et al. have proposed OSIsoft PI system augmented with semantic models [12]. The OSIsoft PI system has targeted DTs to capture, process, analyze, and store real-time operational data within the oil and gas industries. On the other hand, Agniva et al. have introduced a simple mechanism of formalizing knowledge as DT models coming from sensors in industrial production lines [3]. Their proposed mechanism utilizes graph-based query language to extract features from data, infer knowledge from large-scale production line data, and enhance manufacturing process management with reasoning capabilities. In addition, the authors of Reference [13] have introduced a conceptual architecture and model for smart manufacturing, relying on service-based DTs. The proposed architecture has described how automatically DTs compose the corresponding physical processes, sharing some analogies with web service composition. In addition, Aivaliotis et al. have proposed an advanced physics-based modeling approach for predictive maintenance using DTs [14]. The proposed approach has been presented in three phases to reduce the modeling efforts and provide a common framework for the modeling of different resources enabling DTs: (1) machine modeling, including modeling the dynamic behavior of the machine using a black box, grey box, and white box models; (2) virtual sensor modeling, data is gathered during the simulation; (3) modeling parameters definition, the modeling parameters are updated with a suitable frequency to ensure the DT of the machine.

Regarding the role of DTs and data analytics, Song et al. have proposed a model-based machine predictive maintenance that comprises a plurality of DTs, and a simulation platform [15]. The majority of DTs corresponds to the plurality of remotely located physical machines. Each respective DT comprises product nameplate data corresponding to a unique physical device, one or more simulation models, and a database containing run time log data collected from sensors associated with the special physical machine. Furthermore, a modular-based corrective maintenance methodology using DTs has been proposed to automate decision-making in complex systems [16]. The proposed modular corrective maintenance methodology has relied on developing DTs from engineering data using AutomationML.

In Reference [17], the authors discussed the DT and big data in smart manufacturing by focusing on applications, production, manufacturing, maintenance prediction, etc. In contrast, the authors of Reference [18] presented the enabling tools and technologies for DT in smart manufacturing. B. He et al. [19] introduced the DT-driven sustainable smart manufacturing. Furthermore, the authors of Reference [20] focused on DTs with Industry 4.0 and data analytics. Sivalingam et al. [21] discussed the investigation of wind farm used and power consumption for smart manufacturing. They mainly focused on data analytics and IoT device within DT to predict and perform wind turbines maintenance. These studies focused on the importance of DT for smart manufacturing and data analysis. Still, our proposed framework focuses on DTs' collaboration for erratic automatic operation data prediction for Energy 4.0 for wind turbines in smart manufacturing. Furthermore, Table 1 describes a comparison of existing work and the present work concerning the applications, including collaboration-based, DT-bases, Industry 4.0-based, and operational data analysis-based.

**Table 1.** Comparison of existing work and the present work.

| Ref | Highlighted | Applications | | | |
|---|---|---|---|---|---|
| | | Collaboration-Based | DT-Based | Industry 4.0-Based | Operational Data-Based |
| [5] (2019) | The homogeneous and heterogeneous of multi-robot collaboration to perform their complex task in decentralized fashion with blockchain technology | ✓ | X | X | X |
| [6] (2019) | The importance of machine learning of multi-robot cooperation by keeping connectivity and improving mobility in performing tasks. | ✓ | X | X | ✓ |
| [7] (2019) | The collaboration between drone and IoT devices for improving Industry 4.0 applications, such as smart city, smart healthcare. | ✓ | X | ✓ | X |
| [8] (2018) | DT monitoring that for monitoring and development of wind farms. | X | ✓ | X | ✓ |
| [9] (2017) | An approach for identifying the network physical vulnerabilities in Industry 4.0 systems. | X | ✓ | X | ✓ |
| [10] (2014) | AI-based supervisory control and data acquisition method for prediction and fault diagnosis of wind turbines | X | X | X | ✓ |
| [13] (2019) | A conceptual architecture and model for smart manufacturing relying on service-based DTs | X | ✓ | ✓ | X |
| [14] (2019) | Advanced physics-based modeling approach for predictive maintenance using DTs | X | ✓ | ✓ | ✓ |
| [15] (2016) | A model-based machine predictive maintenance -based on DTs and a simulation platform | X | ✓ | ✓ | ✓ |
| [16] (2018) | A modular-based corrective maintenance methodology using DTs to automate decision making in complex systems | X | ✓ | ✓ | ✓ |
| [17] (2018) | Discussion of the DT and big data in smart manufacturing in terms of applications, production, manufacturing, maintenance prediction | X | ✓ | ✓ | ✓ |
| [18] (2019) | A tool and technologies for DT in smart manufacturing | X | ✓ | ✓ | X |
| [19] (2020) | A DT-driven sustainable technique smart manufacturing | X | ✓ | ✓ | X |
| [20] (2020) | Focusing on DTs with Industry 4.0 and data analytics | X | ✓ | ✓ | ✓ |
| [21] (2018) | Investigation wind farm and power consumption for smart manufacturing using IoT and DTs to perform wind turbines maintenance. | X | ✓ | ✓ | ✓ |
| [22] (2020) | Collaboration of drone and IoT to enhance smartness of smart cities applications. | ✓ | X | X | X |
| Our work | A conceptual framework for DTs collaboration to provide an auto-detection of erratic operational data by utilizing the intelligence of operational data in the manufacturing systems. | ✓ | ✓ | ✓ | ✓ |

### 3. Overview of Digital Twins Collaboration

DTs are merging the virtual worlds and real worlds. It is used to describe the detailed presentation of machine, device, robots in the warehouse, production, and process. The DTs' advantages in Industry 4.0 include improving data security and quality, reducing cost, and making faster decision-making. The authors of Reference [23] described DTs as one to one virtual replica of machine, robots, and devices containing data, function, and communication interfaces. The main parts of DTs are physical entity, virtual entity, and information that connect virtual and physical entities [24,25], as shown in Figure 2.

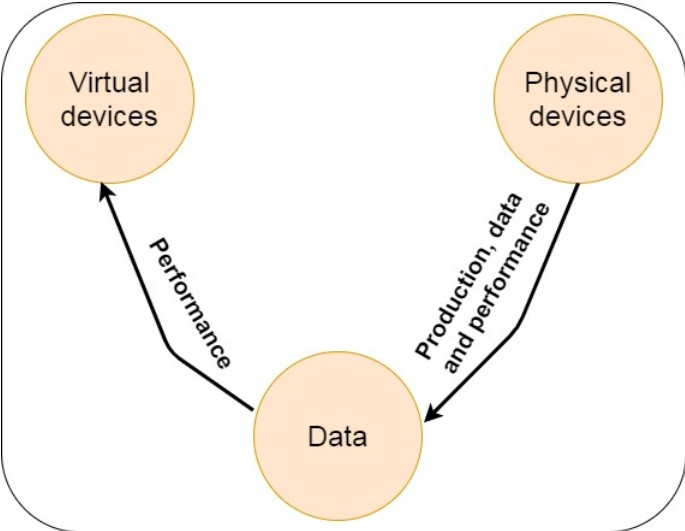

**Figure 2.** The main parts of DTs.

Collaboration means sharing and exchange information among entities and share task to act accordingly. The authors of Reference [6,26] discussed the importance of Artificial Intelligence (AI) and Machine Learning (ML) for robot collaboration based on improving quality of services, connectivity and reliability. Furthermore, collaboration of drone and Internet of things to enhance smartness of smart cities applications [27] and public safety [7], and for better Quality of Service (QoS) [22]. The collaboration among multi-user and identifying the activities is described in Reference [4]. Collaboration of DTs and human described with details in Reference [28]. However, the authors highlighted the challenges of collaboration in industry platform [29–31].

Furthermore, the authors of Reference [5,32] discussed the collaboration among multi-robot to perform common task effectively and efficiently. The Internet of forest aims to detect the forest fire detection discussed [33]. All of the above literature discussed collaboration in smart industries. Still, none of the studies focused on discussing the collaboration of DTs for automatic erratic operational data detection in Industry 4.0, which we address in this paper.

### 4. Proposed Conceptual Framework of Collaborative Digital Twins

In this paper, a conceptual framework of DTs' auto-detection is proposed to enable services, such as fault diagnostics, anomaly detection, and optimization, to understand the DT status, interact with other DTs at the edge level, learn from other DTs, and share common semantic knowledge within automated production system. The proposed framework is considered one level higher than the DT in a cyber-physical production system that could be developed and implemented on top of the DT platform. Furthermore, the conceptual framework for auto-detection of erratic operational data can add value in the context of "plug-and-produce" for Industry 4.0, including predictive maintenance using stored operational data and semantic knowledgebase DT throughout the life-cycle. Six components are required to equip the conceptual framework of DTs with auto-detection intelligence.

As shown in Figure 3, the components being the DT model comprehension, intelligent services, operational data management, knowledgebase management, synchronization, simulation and extra interfaces for communicating with the physical asset are added to make the DTs communicate automatically. These parts will be elaborated flowingly.

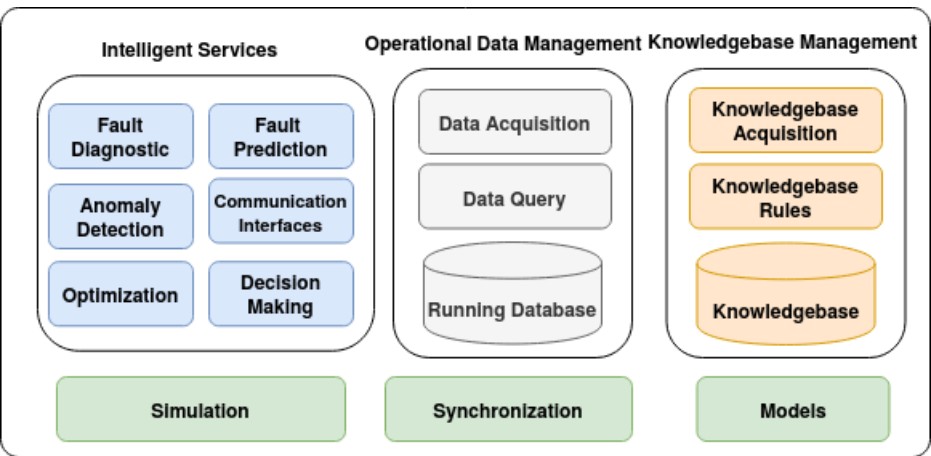

**Figure 3.** The architecture of the conceptual framework for collaborative DTs.

### 4.1. Proposed Framework Model

For representing the DT with the physical asset throughout the entire lifecycle, an element is needed to understand and manage all models and data. For example, Figure 4 shows a DT reference model [34]. At the technical core, the development of DT needs three components: (1) an information model that abstracts the specifications of a physical object; (2) a communication mechanism that transfers bi-directional data between a DT and its physical counterpart; and (3) a data processing module that can extract information from heterogeneous multi-source data to construct the live representation of a physical object. These three components must work together for constructing a DT. Without an information model to abstract a physical entity's features, data transmitted to cyberspace will lose its meaning and context.

Furthermore, the model is used to semantically model the data, reflecting the DT features and their relations using object-oriented concepts. Some semantic work could be done in this module to describe the relationship between the models using model-to-model, e.g., OOP, RDF, and OWL when the complex DT systems have heterogeneous DTs types.

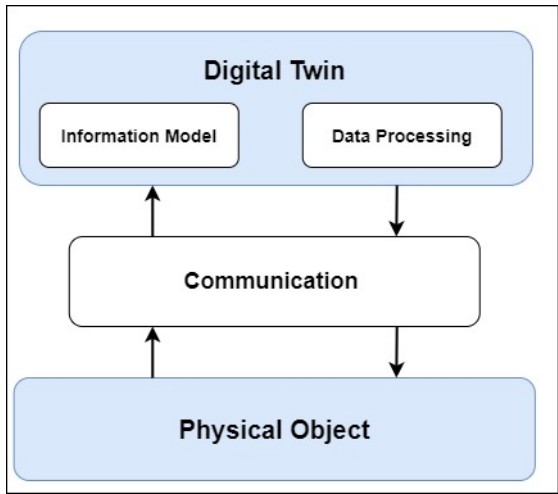

**Figure 4.** A DT reference model [34].

## 4.2. Intelligent Services

Automatic detection for the DT of an asset should include all the functionalities that can identify the anomalies and erroneous data produced from sensors in the real world from various tasks. These services interact intelligently to manage the real product and its virtual counterpart, which are twins to travel a parallel journey from design and development to production and service life. An example of such a service is a robot in the production line, which can cut, drill, glue, etc. DT shows the up-to-date status of the robot. Once the robot changes in the dynamic setting, e.g., the temperature increases in the robot or vibration level is high. These functionalities can identify the problem by analyzing the local operational data at the edge level, e.g., DT level. They can figure out what the problem is and then take appropriate action. For instance, the services can be used within communication with other DTs to make the best decision for abnormal data and warn the decision-makers in case of failure or ignore the erroneous data based on their understanding from other DTs within the machine. Six intelligent services interact together to perform the automatic detection of erratic operational for DT, which are: fault diagnostics, fault prediction, anomaly detection, communication interfaces, optimization and decision-making, and they are described in the following:

### 4.2.1. Fault Diagnostic Service

It is used to detect the new failures (i.e., which potentially happen) by applying knowledge and rules-based analysis.

### 4.2.2. Fault Prediction Service

It is used to predict future failures based on the current machine status by applying machine learning and statistical analysis.

### 4.2.3. Anomaly Detection Service

It is used to identify the rare items, events or observations which raise suspicions by differing significantly from the majority of the data. This brings us to one of the key objectives: *how do we identify whether data points are normal, erroneous or anomalous using machine learning and statistics.*

### 4.2.4. Communication Interfaces Service

It is used to enable automatic access to the physical asset to read the up-to-date data, as well as communicate with other DTs within the operational environment.

### 4.2.5. Optimization Service

It can implement different optimization methods to improve the auto-detection of DT within the virtual world. For example, swarm intelligence (SI) could be employed for decentralized deployed DTs' collective behavior. In regards to DTs data, SI could consist typically of a population of DTs interacting locally with another one and with their environment where the data models of the DTs follow defined rules (i.e., failure rules). For DTs' behavior, there is no centralized control structure dictating how individual DTs should behave. Local and degree of uncertainty should be monitored and studied toward global behavior, which is considered the production system's health conditions.

### 4.2.6. Decision-Making Service

It is used to identify a decision based on the gathering information and assessing alternative solutions within the production system.

## 4.3. Operational Data Management

Operational data management is used to manage up-to-date data in real-time. According to the DT context, operational data management manages the DT data, reflecting the physical, cyber world in real-time. This component consists of three sub-components,

including data acquisition, data query, and running database [35,36]. The data acquisition component is used to collect the sensors data in the real world, which will be injected into the simulated environment parallel to the real world. The data query component is used to retrieve information from the running operational database actively. Running database holds the up-to-date values of the deployed DTs within the virtual cyber system.

### 4.4. Knowledgebase Management

This component is used to manage the knowledgebase within the proposed DT auto-detection framework. It consists of three sub-components, including knowledge acquisition, knowledge rules, and knowledgebase. Knowledge acquisition is a process that is used to define the rules and ontologies required for a knowledgebase [37]. This process describes the initial tasks associated with developing an expert system. In particular, the experts determine the failure rules, the DTs' semantic ontologies, and DTs' hierarchy within the edge level.

Regarding the hierarchy of DTs, each DT relation to other DTs is defined and stored, such as instance-instance relations, inheritance, parent-child relations, and so on, for the whole system to remain consistent. Knowledgebase contains the set of knowledge learned through relevant machine learning techniques from historical maintenance. Knowledge rules are used to refine the knowledgebase by identifying new failures conditions using rule-based analysis.

### 4.5. Simulation

This component is responsible for the simulation of the data exchange among DTs and the entire production system's process flow in the real world. In particular, the DT can extrapolate how an object, process, product, or production system will work in the future after all necessary structural changes are implemented and the potential failures that might happen within the manufacturing process.

### 4.6. Synchronization

The DT should always be in sync with the existing manufacturing cells. The physical asset and its physical relations with other assets within the cyber system (e.g., wiring, physical fixation position, etc.) can be changed very often during its life-cycle. Accordingly, synchronization is implemented as a synchronization interface. It is necessary for synchronizing interdisciplinary models and their relations in a DT to keep a parallel synchronization consistency within the production system.

## 5. Workflow Description

In the context of Industry 4.0, cyber-physical production systems (CPPs) contain different cyber-physical systems (CPSs), which are an integration of digital data and cyber methods with physical processes. These CPSs have their intelligence and communication capabilities. As the DT platform of the CPPS is a composite of many individual DTs of different CPSs, it has significant capabilities regarding the processing of acquired operational data. It can apply appropriate algorithms and semantic technologies to the data to conduct data analysis. For example, to detect and understand the DT status, behavioral analytics can be used to refine the DT models by extracting new knowledge from the data. The extracted knowledge reveals new insights into DTs' behavior, which provides assistance to the manufacturers to optimize their production in various concerns. Consequently, in this work, we proposed an auto-detection conceptual framework of erratic operational data described into a workflow model. The workflow of the proposed framework can be used to notify the decision-makers about the potential failures based on the DTs' operational data leading to an increased quality of the manufacturing process. It is presented in terms of four phases; information extraction, change detection, synchronization, and notification, as described below, where their steps are listed according to Figure 5.

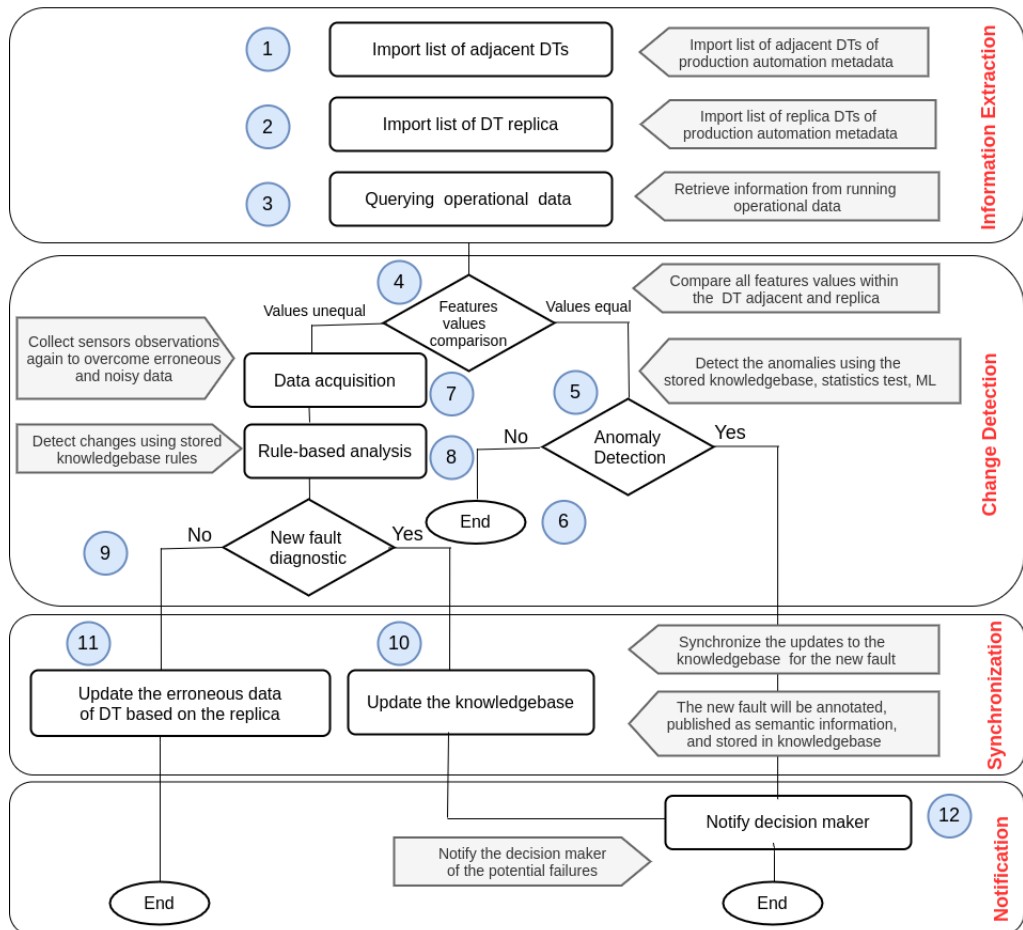

**Figure 5.** The workflow for auto-detection of erratic operational data using collaborative DTs.

### 5.1. Information Extraction

In the phase, the required information (i.e., the features and values) of the deployed DTs, including all adjacent DTs and their replica, are extracted. We describe this phase in three steps: importing the list of adjacent DTs, importing the list of DTs replicas, and querying information [38]. The DTs adjacent list is retrieved from the production system's metadata, which is defined as production system setup. Then, the list of the DT replica is retrieved from the metadata of the production. Finally, the query data component is used to retrieve the information [39]. Furthermore, some semantic can be performed on the base of the more complex DT models to retrieve the required information.

### 5.2. Notification

In this phase, one step is performed. The notification will be sent to the decision-maker in case the anomaly detection service detects the potential failures.

### 5.3. Change Detection

In this phase, we identify the change of the operational data within the DT. Six steps are performed in this phase listed, as shown in Figure 5. The values of the extracted features of the DT and its adjacent and replica are compared. DT adjacents and replica of the extracted features are equal, and then there is a probability that the data are normal or abnormal. Therefore, the data is sent to the anomaly detection service. The anomaly detection service, which implements machine learning techniques and statistics tests using the stored knowledgebase, finds the correlations of operation data with the historical process data of production plants to predict future maintenance. In the case when no anomaly is detected, no notifications will be sent to decision-makers. Furthermore, the simulated virtualized

environment of the production system, i.e., deployed DTs, is considered working in healthy conditions. When all features values of the DT, its adjacents, and replica are unequal. Then, the data acquisition component collects the sensors' data again to overcome the erroneous and noisy data. The rule-based analysis is applied by using the rules as the stored knowledge representation to interpret information from the collected operational data. These knowledgebase rules are used to diagnose and analyse new faults. These rules are sometimes referred to as 'if statements', which depend on the expert system as they tend to follow the line of 'IF X happens, THEN do Y'.

*5.4. Synchronization*

In this phase, we identify the change of the operational data within the DT. Six steps are performed in this phase listed, as shown in Figure 5. In this phase, the new updates are synchronized and stored to make the DT platform consistent. Two steps are performed in this phase listed, as shown in Figure 5. In case the data is diagnosed as a new fault, the new fault will be annotated, published as semantic information, and stored in the knowledgebase. In case the data is not diagnosed as a new fault, it is considered as erroneous data, and then the DT should be updated based on its replica to avoid incorrect observations within the simulated environment.

## 6. Energy 4.0 Fault Diagnosis of Wind Turbines

Fault diagnosis is used to identify the fault cause of the online abnormality, which is crucial for efficient and optimal operation of industrial processes. Due to the time-sensitive industrial applications, the historical fault data may consist of multiple patterns and cannot be described accurately in real-time. So, using a DTs collaboration to estimate the potential fault of the industrial processes in real-time will be useful to detect early faults and take the proper action to decrease downtime within the production systems.

Concerning Energy 4.0, i.e., the wind energy industry, the DTs are used to represent the wind energy system composed of the DTs' representation of the geographical wind farms. The DTs collaboration is used to optimize the operation of the wind energy system and perform early maintenance of physical assets, systems, and production processes to increase the life cycle of their wind systems. The physical asset could be an engine or a turbine that has a set of sensors that can collect real-time data and operational status about the wind system. And, DTs of the wind system are collaborated to track wind farms and identify issues by visualizing the change of the wind system over time.

For a turbine, two main typical elements within a turbine that receive information from sensors are: Programmable Logic Controller (PLC) and Supervisory Control and Data Acquisition (SCADA). PLCs receive information from sensors transforming electrical signals to digital data while SCADA systems are physically connected to the sensors/PLCs collecting signals and other data. For the SCADA system, one turbine could contain 20–30 sensors, and a typical wind farm has 10–100 turbines where zones or geographical regions incorporate 5–50 wind farms. So, it is difficult to detect the fault within the wind system comprising multiple turbines that generate high-frequency data within some tens of milliseconds from geographically wind farms [40]. Furthermore, the generated wind data has other variables, such as weather data, which could affect the wind energy systems.

The DTs system which represents the wind energy system can collaborate to diagnose the fault of the wind system whether caused by hardware failures within a turbine or a farm or weather conditions e.g., wind speed, temperature, humidity. In particular, Figure 6 shows the high-level of the DTs collaboration for fault diagnosis in the wind energy industry where the faulty node could be detected including malfunctioning machines or devices within a farm or a turbine. The DTs collaboration can understand each DT status, interact with other DTs, learn from each other DTs, and share common semantic knowledge across geographically wind farms. For weather conditions, each wind farm which is located in different stations gives weather data including wind speed, temperature, and humidity at predefined time intervals, such as 10 min. There is a strong cross-correlation between the

weather data and the failure data which cause turbulence according to wind farm locations geographically. For example, wind speed turbulence could cause vibration and resonance, high wind speeds could lead to brake failure, low temperatures could leak to brittleness or lubricant freezing, and temperature variations could cause expansions and contractions. So, to keep tracking the weather data within the wind energy system, the DT for the wind energy system represents the weather data of a wind farm. The DTs are collaborated to link these real-time digital weather data to online detect the turbulence of the wind energy system. It allows interaction with other DTs within other wind farms and sharing common semantic knowledge to indicate how weather conditions may affect offshore wind turbine(s) failure rates and increase the downtime of the wind energy system.

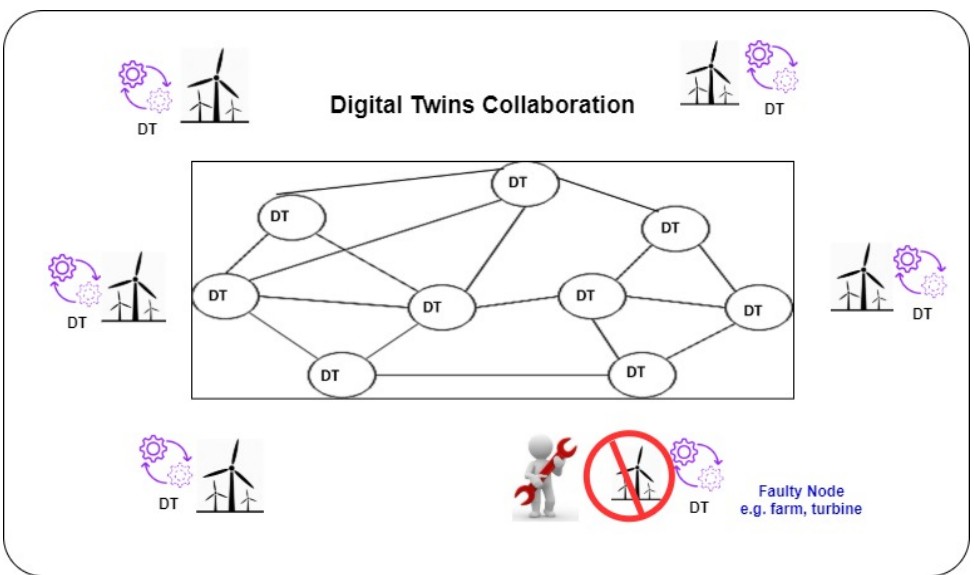

**Figure 6.** DTs collaboration for Energy 4.0 fault diagnosis of wind tribunes.

For the hardware failures, the proposed framework for DTs collaboration can detect the DTs that contain the faulty operational data and locate the sensors within a specific turbine in a particular wind farm that the human eye would miss. For example, the collaborative DTs can compare the power generated by each turbine for that moment (i.e., represented in LV ActivePower (kW) value within a DT) with other adjacent turbines. Considering the homogeneity of turbine manufacturing and the same weather conditions within a farm, the collaborative DTs can detect the faulty turbine(s) or accurately predict the potential failure of a turbine(s) that generate lower power. Furthermore, if the collaborative DTs detect a new fault, the newly diagnosed fault will be annotated, published as semantic information, and stored in the distributed knowledgebase accessible by other DTs within the digital wind energy system. Furthermore, wind farm operators could use the proposed framework based on DT collaboration to predict what might happen if they increase their turbines' power rate or what might happen for the power rate of their turbines under the forecasted weather conditions.

Generally, the DT model is constructed by leveraging physics-based models and data-driven analytics for fault diagnosis. The collaborative DT model is defined based on the basic DT model, and it is composed of three major components, including digital model, data analytics, and knowledgebase [34]. These components are integrated to investigate the DTs collaboration ability of localization for fault diagnosis within the wind energy system. The components includes the digital model, data analysis, and knowledgebase.

The digital model of the operational data contains semi-structured content, e.g., JSON and XML. According to the wind energy industry domain, the data is generated from turbines or weather sensors. The wind turbine data describes the turbine's physical system, and the weather sensors are used to monitor climate change. Their data represent the weather variables, including wind speed, temperature, humidity, CO, and $CO_2$. Data

analytics supports health analyses and maintenance decision-making by leveraging digital simulation and data-driven intelligence. For instance, data analytics can be used within DTs' interaction and communication to describe, diagnose, predict, and prescribe the physical wind system's behavior for fault diagnosis. The data analytics outcome will be used as inputs to notify the decision-makers to make the best decision for abnormal data in case of potential failure over the wind energy industry system. Furthermore, knowledgebase contains the set of knowledge learned through relevant machine learning techniques from historical maintenance.

## 7. Discussion and Future Direction

Additional semantic capabilities (e.g., knowledge graphs) could be explored to represent complex cyber-physical production systems. Furthermore, the DT model is used to semantically modeling the data, reflecting the DT features and their relations using object-oriented concepts. Some semantic work could be done to describe the relationships between the DTs models using model-to-model, e.g., OOP, RDF, and OWL in the complex DT systems heterogeneous DTs types.

Task allocation issues also arises as machine or robot, and human, can achieve the task. Therefore, simulation can allocate and distribute the tasks between robot and human-based on availability and resource. Thus, the DTs, which represent human and robot, can collaborate efficiently in smart manufacturing to achieve tasks accordingly. For flexibility and productivity, DTs should be used to balance and increase processing during work. For any change in the task execution, the robot must take a new trajectory to reach the goal and achieve the tasks. Therefore, collaboration between the DTs representing human and robot in smart manufacturing can help each other accomplish the required tasks effectively and efficiently.

Furthermore, blockchain technology could be combined with DTs collaboration to guarantee transparency, decentralized data storage, data sharing, peer-to-peer communication, secure and trusted traceability, and scalability.

## 8. Conclusions

In this research work, we presented a conceptual framework for DTs collaboration to auto-detect the production system's erratic operational. The DTs collaboration can detect failures and erratic sensors, device, and annotate them on the DT platform using semantic capabilities, such as knowledgebase and knowledgebase rules. The proposed DTs collaboration-based framework also provides an interaction mechanism to understand the DT status, interact with other DTs, learn from other DTs, and share common knowledge. Furthermore, the interaction mechanism can detect the anomalies at the edge level and understand the operational environments' overall picture and conditions. A use case of Energy 4.0 fault diagnosis for wind turbines is described to discuss how the proposed auto-detection framework using the DTs collaboration can help identify and diagnose the potential failure, e.g., malfunctioning nodes within the energy industry.

**Author Contributions:** R.S.: Conceptualization, Data curation, Formal analysis, Methodology, Writing—Original draft, Writing—Review & editing. S.H.A.: Conceptualization, Methodology, Formal analysis, Writing—Review & editing. J.G.B.: Conceptualization, Funding acquisition, Investigation, Project administration, Brown, K.N.B.: Conceptualization, Funding acquisition, Investigation, Project administration. M.I.A.: Conceptualization, Funding acquisition, Project administration, Supervision. All authors have read and agreed to the published version of the manuscript.

**Funding:** This research has emanated from research supported by a research grant from Science Foundation Ireland (SFI) under Grant Number SFI/16/RC/3918 (Confirm), and Marie SkłodowskaCurie grant agreement No. 847577 co-funded by the European Regional Development Fund.

**Institutional Review Board Statement:** Not applicable.

**Informed Consent Statement:** Not applicable.

**Data Availability Statement:** No Data available online. For further query email to corresponding author (radhya.sahal@nuigalway.ie, radhya.sahal.dsi@gmail.com).

**Acknowledgments:** This research has emanated from research supported by a research grant from Science Foundation Ireland (SFI) under Grant Number SFI/16/RC/3918 (Confirm), and Marie Skłodowska-Curie grant agreement No. 847577 co-funded by the European Regional Development Fund.

**Conflicts of Interest:** The authors declare no conflict of interest.

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
