# Peer review of "Digital Twins Collaboration for Automatic Erratic Operational Data Detection in Industry 4.0"

_applsci, doi:10.3390/app11073186_

Round 1

Reviewer 1 Report

The subject of the review is the article entitled „Digital Twins Collaboration for Automatic Erratic Operational Data Detection in Industry 4.0”.

In the introduction, the authors explain the key concept of DT. The use of this phrase seems most appropriate, considering the wide range of features it represents. The subject of DT itself may not be something very new, because for years virtual models of real elements have been created in a computer environment to verify and test them with the use of many different parameters. However, the authors point to the new possibilities and advantages of using DT in industry. The authors place particular emphasis on the use of DT in the industry of obtaining energy from renewable sources - they focus on turbines and wind farms. The article is not a typical literature review, because the authors not only summarize the available knowledge, but also pose specific questions that they try to answer and additionally present their own concept of using DT in forecasting and diagnostics in the field of wind energy industry.

In the literature part, the authors cite source texts, on the basis of which they explain the entire concept of smart manufacturing, Industry 4.0, wind farms and, in general, the possibilities of using DT. Then the authors present the architecture of their original concept, in which they indicate the elements necessary in this model. Their versatility indicates a good orientation of the authors in the issues of modeling, and in addition, the multitude of these elements may indicate the effectiveness of the model in forecasting phenomena disturbing the functionality of devices in real-time.

The content-related correctness of the article does not raise any objections. The article is well written, the chapters and subsections are systematized, thanks to which - despite the volume of content, it is easy to follow the authors' thinking. In the part that presents the authors' concept, however, there are no diagrams, charts, photos that would complete the content and allow the reader to visualize what was described in words (I believe that the two drawings on page 13 do not reflect the entire concept).

Question: at the end of the work, after figures 5 and 6, there are two blank pages. Were there any sticky elements on these pages that are not visible on the copy sent for review?

In summary, the article is an interesting, well-written and organized. Taking into account the sources of financing, it can also be stated that the subject of the article taken up by the authors is part of a broader strategic research, the results of which may be important for the entire industry. Nevertheless, due to the specificity of the subject and the article itself, I am not sure if the choice of the journal - Applied Science - is right. Perhaps the list of MDPI journals is different, thematically more consistent with the topic proposed by the authors.

Author Response

Question: at the end of the work, after figures 5 and 6, there are two blank pages. Were there any
sticky elements on these pages that are not visible on the copy sent for review?

Authors Response: 
Thank you for your invaluable comment.

Actions:
There is no element there just was a formatting issue and we have fixed it.

Reviewer 2 Report

Overall, this paper is interesting, well structured and self-contained. I think that the contribution herein introduced may have several and useful practical implications. The paper appears to be sound and relies on well-know building blocks. Again, it is written by using an appropriate technical language. 

I want suggest this paper to authors:

- Colace, F., De Santo, M., Lemma, S., Lombardi, M., Pascale, F.: BotWheels: a petri net based chatbot for recommending tires. In: The 3rd Edition of Special Session on Knowledge Discovery Meets Information Systems: Applications of Big Data Analytics and BI - Methodologies, Techniques and Tools (KomIS), within the 6th International Conference on Data Science, Technology and Applications (DATA) (2017)

As minor remarks:
The paper should be better compared and contextualized with respect to the state of the art.
In some parts of the paper, the clarity and editorial quality of the paper weaken. As a consequence, such parts result to be quite difficult to read. Therefore, I would suggest to carefully improve the prose of writing in order to make this paper easier to read.
The quality of the figures must be improved.
An accurate proofreading is strongly recommended.

Author Response

Could you please find the attached file? 

Reviewer 3 Report

The paper is well prepared. I found some problems in the text, but they do not they do not lower the substantive value of the text:

  • page 2, line 22: You have "[2];" - I think you should use ":", because then you list elements
  • page 3, line 75: you should add ":" on the end of the line
  • page 3, line 95: IoT - I think you should write the full name (even in brackets), for some readers it can be helpful
  • page 3, line 100: moreover - you should start with capital letter. It is new sentence
  • page 4, line 132:  simulation 3) modeling  - you should add "," after simulation
  • page 4, line 166: QoS - like IoT, you should write full name in brackets
  • page 8, line 305: production. finally - finally should start with capital letter
  • page 9, line 351: "sensors are;" - I think you should use ":"
  • page 10, line 397: "analysis,and knowledgebase" - you should remove ","
  • page 11, lines 449, 463, 471, 474, 477: you should give year of publication in bold
  • page 12, line 481: you should give year of publication in bold
  • bibliography: no 20 & 28 - please, add number of pages

Author Response

(The authors gave the same response as above.)
